# Machine Learning for Lung Cancer Subtype Classification: Combining Clinical, Histopathological, and Biophysical Features

**DOI:** 10.3390/diagnostics15020127

**Published:** 2025-01-07

**Authors:** Aiga Andrijanova, Lasma Bugovecka, Sergejs Isajevs, Donats Erts, Uldis Malinovskis, Andis Liepins

**Affiliations:** 1SIA “APPLY”, Ieriku Street 5, LV-1084 Riga, Latvia; andis.liepins@applyit.lv; 2Institute of Chemical Physics, Faculty of Science and Technology, University of Latvia, Jelgavas Street 1, LV-1004 Riga, Latvia; lasma.bugovecka@lu.lv (L.B.); donats.erts@lu.lv (D.E.); uldis.malinovskis@lu.lv (U.M.); 3Faculty of Medicine and Life Sciences, University of Latvia, Jelgavas Street 3, LV-1004 Riga, Latvia; sergejs.isajevs@lu.lv

**Keywords:** lung cancer, cancer subtype classification, atomic force microscopy, biophysical properties, machine learning, Bayesian networks, personalized medicine, cancer biomarkers, cell mechanics, non-small cell lung cancer

## Abstract

**Background/Objectives:** Despite advances in diagnostic techniques, accurate classification of lung cancer subtypes remains crucial for treatment planning. Traditional methods like genomic studies face limitations such as high cost and complexity. This study investigates whether integrating atomic force microscopy (AFM) measurements with conventional clinical and histopathological data can improve lung cancer subtype classification. **Methods:** We developed and analyzed a novel dataset combining clinical, histopathological, and AFM-derived biophysical characteristics from 37 lung cancer patients. Various machine learning techniques were evaluated, with a focus on Bayesian Networks due to their ability to handle complex data with missing values. Leave-One-Out Cross-Validation was employed to assess model performance. **Results:** The integration of biophysical features improved classification accuracy from 86.49% to 89.19% using a data-driven Bayesian Network model, though this improvement was not statistically significant (*p* = 1.0). Four key features were identified as highly predictive: sex, vascular invasion, perineural invasion, and ALK mutation. A simplified model using only these features achieved identical performance with significantly reduced complexity (BIC 51.931 vs. 268.586). **Conclusions:** While AFM-derived measurements showed promise for enhancing lung cancer subtype classification, larger datasets are needed to fully validate their impact. Our findings demonstrate the feasibility of incorporating biophysical measurements into cancer classification frameworks and identify the most predictive features for accurate diagnosis. Further research with expanded datasets is needed to validate these findings.

## 1. Introduction

Lung cancer remains one of the most common cancers worldwide with high mortality rates, posing significant challenges to public health and medical research. Lung cancer is the most commonly diagnosed cancer worldwide (12.4% of the total cases). According to GLOBOCAN in 2022 there were 2480301 new cases of lung cancer, and 1817172 mortality deaths cases [1]. Accurate classification of lung cancer subtypes is crucial for effective patient follow-up and treatment planning. While genomic studies have provided valuable insights into lung cancer biology, they often face limitations such as high cost, complexity in data interpretation, and challenges in clinical implementation. Histopathological examinations remain the gold standard for lung cancer diagnosis. However, the proper classification of poorly differentiated lung cancer could be challenging, regardless of immunohistochemical and genetical testing.

Several studies have explored machine learning approaches for lung cancer subtype classification, utilizing various data types and methodologies. Liu et al. analyzed a dataset of 349 patients using 1029 CT radiomics features, achieving 86% accuracy with Support Vector Machines (SVM) and feature selection [2]. Sha et al. combined radiomics features from Positron Emission Tomography (PET) images with clinical features in a dataset of 100 patients, achieving an AUC of 0.781 when integrating both radiomics and clinical features [3]. Shen et al. took a more comprehensive approach, incorporating clinical data, PET Standardized Uptake Values, and 385 PET/CT radiomics features from 250 patients, with SVM achieving an accuracy of 86.23% [4].

Recent advances in biophysical measurement techniques have enabled the quantification of mechanical properties of individual cancer cells with high precision [5]. Atomic Force Microscopy (AFM) has emerged as a powerful tool in this field, offering the ability to measure cellular elasticity, surface roughness, and dimensions at the nanoscale [6]. These biophysical properties have shown promise as potential biomarkers for cancer detection and classification [7,8]. Cross et al. and Plodinec et al. demonstrated that cancer cells generally exhibit lower stiffness compared to normal cells, though their studies focused primarily on breast cancer.

More recent studies have continued to explore various approaches to lung cancer classification. Wang et al. employed a combination of clinical, laboratory, and CT radiomics features in a study of 138 patients, achieving an AUC of 0.995 with random forest models [9]. Zhao et al. integrated clinical, laboratory, and PET/CT radiomics features from 120 patients, with their SVM model achieving 80% accuracy in distinguishing between lung cancer subtypes [10].

To provide context for our study’s methodology and results, Table 1 and Table 2 present a comprehensive overview of recent machine learning approaches to NSCLC subtype classification.

Our study differs from previous work in several key aspects:Integration of AFM measurements: We introduce novel biophysical features derived from atomic force microscopy, providing direct mechanical characterization of cancer cells.Probabilistic approach: While previous studies primarily used deterministic classifiers, we employ Bayesian Networks to handle uncertainty and missing data inherent in clinical settings.Feature interpretability: Our approach identifies a minimal set of highly predictive features (sex, vascular invasion, perineural invasion, and ALK mutation) while maintaining classification performance.Comprehensive validation: We validate our methods using both the novel LCPP dataset and the well-established WDBC dataset, demonstrating the robustness of our approach.

While these studies demonstrate the potential of machine learning in lung cancer classification, the integration of AFM-derived biophysical measurements with conventional clinical and histopathological data remains largely unexplored. This gap presents an opportunity for novel research combining these diverse data types. Furthermore, while most previous studies have used traditional machine learning methods, the potential of probabilistic approaches like Bayesian Networks, which can handle complex, multidimensional data while providing interpretable results, has not been fully investigated in this context [11].

This study aims to investigate whether the integration of AFM-derived biophysical measurements with conventional clinical and histopathological data can enhance the accuracy of lung cancer subtype classification using machine learning techniques. We utilize a novel dataset, the Lung Cancer Physical Properties (LCPP) dataset, which combines the clinical, histopathological, and biophysical characteristics of 37 lung cancer patients. Our research focuses on distinguishing between two major subtypes of non-small cell lung cancer: adenocarcinoma (ADC) and squamous cell carcinoma (SCC) [12].

We employ various machine learning techniques, with a particular focus on Bayesian Networks due to their ability to handle complex, multidimensional data and their interpretability in clinical contexts. Our study not only explores the potential of biophysical features in improving classification accuracy but also investigates the relationships between these novel measurements and conventional clinical indicators.

The findings of this study contribute to the growing field of biophysical oncology by demonstrating the feasibility of incorporating AFM measurements into lung cancer subtype classification. By combining conventional clinical data with novel biophysical measurements and employing advanced machine learning techniques, we aim to contribute to the improvement of lung cancer subtype classification and explore the potential of cell mechanical properties in enhancing cancer diagnostics [13].

The primary objectives of this study were:To evaluate whether integrating AFM-derived biophysical measurements with conventional clinical and histopathological data can improve lung cancer subtype classification accuracyTo identify the most predictive features for lung cancer subtype classification through systematic feature selectionTo assess the effectiveness of probabilistic models, particularly Bayesian Networks, in handling complex, multidimensional medical data with missing valuesTo develop an interpretable classification framework suitable for clinical decision support

These objectives address current gaps in lung cancer diagnostics by:Exploring novel biophysical biomarkers that could complement existing diagnostic methodsDeveloping methods to handle the practical challenges of incomplete clinical dataCreating interpretable models that can support clinical decision-makingIdentifying minimal sets of predictive features to optimize diagnostic efficiency

## 2. Materials and Methods

The methodology was specifically designed to address each research objective:To evaluate biophysical measurements’ impact:-Systematic comparison of models with and without AFM features-Statistical significance testing of performance differences-Analysis of feature interactions through Bayesian NetworksTo identify predictive features:-Implementation of multiple feature selection methods-Cross-validation to ensure robustness-Comparison with domain expert knowledgeTo assess probabilistic models:-Comparison of learned vs. expert-defined structures-Evaluation using multiple performance metrics-Analysis of model uncertainty and confidenceTo ensure clinical interpretability:-Focus on transparent model architectures-Validation against clinical knowledge-Feature importance analysis

### 2.1. Dataset Description

The Lung Cancer Physical Properties (LCPP) dataset comprises clinical, histopathological, and biophysical characteristics of 37 lung cancer patients. The dataset encompasses 23 features for each patient: 3 demographic and clinical features, 10 histopathological features, 4 genetic biomarkers, and 6 biophysical measurements.

The dataset includes patients with two NSCLC subtypes—squamous cell carcinoma (SCC) and adenocarcinoma (ADC), which together account for approximately 85% of all lung cancer cases. The tumour classification and grading was performed according to new WHO Classification of Thoracic tumours [14]. Cancer staging follows the TNM classification system, with stages IA1, IA2, IA3, IB, IIA, and IIB classified as early-stage cancers, and stages IIIA, IIIB, and IIIC as late-stage cancers.

The histopathological slides stained with hematozylins eosin and immunohistochemical were analyzed. The tumour grade, lymphovascular invasion (LVI), perineural invasion (Pn) was assessed. Immunohistochemically PD-L1, ALK and ROS-1 expression was assessed.

PD-L1 IHC 22C3 pharmDx (Agilent) test was used to evaluate PD-L1 expression, which is a qualitative immunohistochemical assay using monoclonal Mouse Anti-PD-L1, Clone 22C3 for use in the detection of PD-L1 protein in formalin-fixed, paraffin-embedded (FFPE) non-small cell lung cancer (NSCLC). PD-L1 protein expression was determined by using Tumor Proportion Score (TPS), which is the percentage of viable tumor cells showing partial or complete membrane staining at any intensity. The specimen was defined as positive if TPS ≥ 1.0%.

For ROS-1 assessment, VENTANA ROS1 (clone SP384) rabbit monoclonal antibody was used. ROS1 immunohistochemistry staining results were interpreted by using four previously described criteria: (1) an H-score with a threshold for ROS1 positivity defined as at least 100; (2) an H-score cutoff of at least 150; (3) an intensity criterion with the cutoff of positivity defined as 2+ or higher in any tumor cells; (4) a positive status based on an intensity of 2+ or higher in at least 30% of the total tumor cells [15].

For ALK assessment, VENTANA ALK (D5F3) CDx Assay was used. The tumour was defined ALK positive if immunohistochemical strong granular cytoplasmic staining in tumor cells (any percentage of positive tumor cells) was observed.

NTRK expression was assessed by VENTANA pan-TRK (clone EPR17341) immunohistochemical assay. Pan-TRK expression was considered positive if the following subcellular staining patterns of any intensity were observed in ≥1% of tumor cells. Cytoplasmic and nuclear reactivity was considered positive for expression. Punctate staining alone was considered non-specific or equivocal.

For EGFR testing, the Biocartis Idylla^TM^ EGFR Mutation Test was performed on the Biocartis Idylla^TM^ system.

Biophysical measurements have varying levels of completeness:51.4% of patients have tumor cell height and width measurements40.5% have elasticity measurements64.9% have surface roughness measurements

Only 27.0% of patients have a complete set of biophysical measurements. Analysis of missing data patterns indicates that data is not Missing Completely at Random (MCAR), as significant relationships were found between the missingness of certain variables and the values of other variables in the dataset.

The missingness in elasticity measurements was found to be significantly related to lymphatic vessel invasion, vascular invasion, and cancer stage. Additionally, the missingness in surface roughness measurements was significantly associated with NTRK mutation status and cell dimension measurements. These relationships suggest that the data is either Missing at Random (MAR) or Missing Not at Random (MNAR) (Table 3).

### 2.2. Data Collection

Patient recruitment and clinical data collection were conducted by the Department of Pathology, Faculty of Medicine and Life Sciences at the University of Latvia. The study protocol was approved by the Faculty of Medicine and Life Sciences, University of Latvia Ethics Committee (Number. 19-25/164). The study was conducted according to the declaration of the Helsinki and Oviedo conventions. All patients signed informed consent to participate in the study.

Lung biopsy section samples were examined using light microscopy analysis according to the current World Health Organization Classification of Thoracic Tumors and College of American Pathologists (CAP) guidelines. PD-L1, ALK, ROS, and NTRK biomarkers expression was assessed by immunohistochemistry. The EGFR gene mutation was assessed by polymerase chain reaction.

### 2.3. AFM Instrumentation

AFM measurements were performed using:Atomic force microscope MFP-3D (Asylum Research, Goleta, CA, USA) with maximum scanning area 90 × 90 $\mu m^{2}$, X and Y sensor sensitivity < 0.5 nm, Z range > 15 μm, Z sensor sensitivity < 0.25 nmThree types of AFM probes:-AC160TS (Olympus, Tokyo, Japan): cantilever size 160 × 40 × 3.7 μm, spring constant 26 N/m, resonance frequency 300 kHz-AC240TS (Olympus): cantilever size 240 × 40 × 2.3 μm, spring constant 2 N/m, resonance frequency 70 kHz-MSCT-AUHW (Park Scientific Instruments, Sunnyvale, CA, USA): cantilever size 310 × 20 μm, spring constant 0.01 N/m, resonance frequency 7 kHzVideo monitoring system CV-S3200 (JAI Corporation, Mumbai, India) with 752 × 582 pixel resolutionFiber-Lite MI-150R illumination sourceTS-140 vibration isolation table (Table Stable LTD, HERZAN, Laguna Hills, CA, USA)

### 2.4. AFM Measurement Protocols

Figure 1 demonstrates the AFM measurement methodology used for tissue section analysis. AFM height images of lung adenocarcinoma (Figure 1a–c) and squamous cell carcinoma (Figure 1d–f) tissue sections are shown at different magnifications. The initial 40 × 40 $\mu m^{2}$ scans (Figure 1a,d) provide an overview of the tissue architecture, while higher resolution 5 × 5 $\mu m^{2}$ scans (Figure 1b,e) reveal detailed surface features.

AFM measurements were conducted on both liquid-based cytology and biopsy section samples to obtain three main categories of biophysical data:

#### 2.4.1. Cell Elasticity Measurements

Quantified as Young’s modulus using liquid-based cytology samplesForce-displacement curves generated through AFM nanoindentation experimentsAnalysis performed using the Hertz contact mechanics modelMeasurements taken separately for cytoplasm and nucleus regions1–4 distinct cells analyzed per patient with 1–10 measurement points per regionParameters used for Hertz model:-Spherical tip geometry (60 nm radius)-Sample Poisson’s ratio = 0.5-Silicon nitride tip properties: Poisson’s ratio = 0.25, E = 290 GPa

Figure 2 illustrates the process of force measurements on individual cancer cells. The AFM height image (Figure 2a) shows an individual cell with marked measurement points (P0–P7) where force curves were obtained. The optical image (Figure 2b) demonstrates the positioning of the AFM probe relative to the cells. The resulting force-displacement curves (Figure 2c) from both cytoplasm and nucleus regions, along with the calibration curve, show distinct mechanical responses from different cellular regions.

#### 2.4.2. Surface Roughness Analysis

Performed on biopsy section samplesInitial scan of 60 × 60 $\mu m^{2}$ area followed by detailed scans down to 2 × 2 $\mu m^{2}$1–3 samples analyzed per patient3–27 measurement lines per sampleRMS roughness calculated from vertical cantilever deflections

#### 2.4.3. Cell Dimension Measurements

Performed on cytology smears through topographical data analysisMeasurements included cell width, cytoplasmic region height, and nuclear region height1–5 cells measured per patientThree measurements taken for each dimension per cellAnalysis focused on maximum vertical distance between substrate and cell’s highest point

### 2.5. Data Analysis

All AFM data were processed using Igor Pro software (versions 6.37 and 6.38B01) with MFP 3D extensions (versions 16.26.227 and 16.52.236). For elasticity calculations, the Hertz model was applied to force-displacement curves. Surface roughness was quantified as root mean square (RMS) of height deviations. For cell dimensions, measurements were averaged at both the cellular and patient levels.

### 2.6. Data Preprocessing

For AFM measurements, which contained multiple observations per patient, the following aggregation strategy was employed:Surface roughness: mean RMS value calculated from 3–27 measurement lines per sample, then averaged across 1–3 samples per patientCell dimensions: three measurements per dimension (height, width) averaged for each cell, then averaged across 1–5 cells per patientElasticity: mean values calculated from 1–10 measurement points per region (cytoplasm/nucleus), then averaged across 1–4 cells per patient

All continuous variables were tested for normality using the Shapiro-Wilk test. Pack-years smoked was the only variable potentially showing normal distribution (*p* < 0.05), though visual inspection did not confirm this.

For traditional machine model training, continuous features were standardized, and categorical features were transformed using one-hot encoding. Binary features remained unaltered. Due to the presence of missing values in biophysical features, these were excluded from traditional machine learning models but included in Bayesian Network analyses.

For Bayesian Network models, continuous variables were discretized into four categories (Low, Medium-Low, Medium-High, and High) using percentile-based binning.

### 2.7. Machine Learning Models

Our model selection process was guided by three key considerations:Small sample size: Given our limited dataset (n = 37), we prioritized models known to perform well with small datasets.Missing data handling: Models needed to effectively handle missing values in biophysical measurements.Interpretability: Selected models needed to provide interpretable results for clinical applications.

Several machine learning models were implemented and compared:Logistic RegressionSupport Vector Machines (SVM) with linear, RBF, and polynomial kernelsDecision TreesK-Nearest Neighbors (KNN)Naive BayesGaussian Process ClassifierLinear Discriminant Analysis (LDA)

For the traditional machine learning models, we employed a systematic grid search approach for hyperparameter optimization.

### 2.8. Bayesian Networks

Two Bayesian Network structures were evaluated: an inferred structure based on domain knowledge and a learned structure derived from the data. The hill-climbing search algorithm with the K2 score was employed for structure learning. Maximum Likelihood Estimation was used for parameter learning, and the variable elimination algorithm was employed for probabilistic inference.

### 2.9. Model Evaluation

Due to the small sample size of the LCPP dataset, Leave-One-Out Cross-Validation (LOOCV) was used for all experiments. Performance metrics included accuracy, precision, sensitivity, specificity, F1-score, and Area Under the Receiver Operating Characteristic curve (AUC-ROC).

Additionally, we calculated Cohen’s Kappa (κ) and Matthews Correlation Coefficient (MCC) to assess model reliability and account for class imbalance:(1)$$\kappa =\frac{p_{o}-p_{e}}{1-p_{e}}$$
where $p_{o}$ is the observed agreement and $p_{e}$ is the expected agreement by chance, and(2)$$MCC=\frac{TP\times TN-FP\times FN}{\sqrt{\left(TP+FP)(TP+FN)(TN+FP)(TN+FN\right)}}$$
where TP, TN, FP, and FN represent True Positives, True Negatives, False Positives, and False Negatives respectively.

### 2.10. Statistical Analysis

Statistical analysis was performed using Python (version 3.10.12) with pandas (version 2.2.2) for data manipulation and SciPy (version 1.13.1) for statistical tests. Mann-Whitney U test was used to compare continuous variables between groups. Chi-square test was used for categorical variables. Correlations were assessed using Spearman’s rank correlation coefficient. *p*-values < 0.05 were considered statistically significant.

McNemar’s test was employed to assess the statistical significance of differences in accuracy between models. DeLong’s test was used to evaluate the statistical significance of differences in AUC-ROC between models.

Power analysis was conducted to assess the statistical validity of our findings. For elasticity measurements between early and late stage samples, we found a large effect size (Cohen’s d = 6.63) with 100% statistical power at our current sample size (n = 37), well exceeding the required sample size of 20 for achieving 80% power. However, for detecting differences in classification accuracy with and without biophysical features (89.19% vs. 86.49%), the analysis revealed low statistical power (6%), suggesting that a larger sample size would be needed to conclusively demonstrate the impact of biophysical features on classification performance.

## 3. Results

### 3.1. Patient Demographics and Clinical Characteristics

The dataset comprises 37 patients, with 17 diagnosed with adenocarcinoma (ADC) and 20 with squamous cell carcinoma (SCC). 17 patients with invasive nonmucinous adenocarcinoma were enrolled in the study. 3 patients had Grade 1 lepidic adenocarcinoma, 5 patients had Grade 2 acinar adenocarcinoma, 4 patients had Grade 2 papilary adenocarcinoma, 2 patients had Grade 3 solid adenocarcinoma, 1 patients had Grade 3 micropapillary adenocarcinoma, 2 patients had Grade 3 acinar adenocarcinoma. The ADC cohort showed a relatively balanced distribution across both cancer stage and sex. Specifically, 7 patients presented with early-stage disease while 10 exhibited late-stage progression. The sex distribution was nearly equal, with 8 male and 9 female patients. Figure 3 illustrates the key demographic and clinical characteristics of our study cohort.

In contrast, the SCC group demonstrated notable demographic skews. There was a pronounced sex imbalance, with 19 male patients (95%) and only 1 female patient (5%). This observation aligns with previous findings indicating higher SCC prevalence in males. A chi-square test revealed a significant association between cancer subtype and gender ($\chi^{2}=8.416$, p=0.004). The stage distribution in the SCC group favored early-stage disease, with 12 patients (60%) classified as early-stage compared to 8 late-stage cases (40%).

The patient cohort spanned an age range of 55 to 75 years. ADC patients had a significantly lower mean age (65.24±4.56 years) compared to SCC patients (68.45±4.61 years) (Mann-Whitney U test, p<0.01). This age distribution aligns with previous studies showing that ADC tends to occur in younger patients.

Smoking history, quantified in pack-years, also showed significant differences between cancer subtypes. SCC patients reported a higher mean smoking history (26.5±6.61 pack-years) compared to ADC patients (21.35±7.00 pack-years) (Mann-Whitney U test, p<0.01). This observation supports previous findings of stronger associations between smoking behavior and SCC development.

Cancer grade distribution varied between subtypes. ADC cases showed heterogeneous grade distribution across both early and late stages. In contrast, SCC cases were predominantly moderately or poorly differentiated, with no well-differentiated (grade 1) cases observed. Late-stage cancers had larger mean size (4.9±1.6 cm) compared to early-stage cancers (3.4±1.6 cm), though this difference was not statistically significant.

### 3.2. Histopathological Features Analysis

Analysis of tumor invasion pathways revealed clear associations with disease stage. Vascular invasion (VI) was observed in 88.9% of late-stage patients compared to 15.8% of early-stage patients. Similarly, lymphatic vessel invasion (LVI) was present in all late-stage patients and 73.7% of early-stage patients. Perineural invasion (PNI) was observed in only 3 patients (8.1%), all of whom had SCC and late-stage cancer.

PD-L1 expression levels showed considerable variation within each subgroup. Late-stage ADC patients demonstrated the highest mean PD-L1 expression (37.40%±27.41%), while early-stage ADC patients showed the lowest (24.86%±25.48%). SCC patients exhibited an inverse pattern, with higher PD-L1 expression in early-stage (35.75%±36.73%) compared to late-stage (25.75%±32.33%) disease. The substantial variability within each group, indicated by large standard deviations, suggests caution in drawing definitive conclusions.

FOXP3 expression patterns varied significantly between cancer subtypes and stages. In ADC cases, FOXP3 expression increased markedly from early to late stages (early-stage: 19.29±10.37 cells/mm^2^; late-stage: 37.70±32.91 cells/mm^2^). SCC cases showed a different pattern, with higher expression in early-stage (33.00±36.03 cells/mm^2^) compared to late-stage (30.50±27.20 cells/mm^2^) disease.

Genetic biomarker analysis revealed several notable patterns. EGFR mutations were present in 16 patients (43.2%), with higher prevalence in ADC (10 patients, 58.8%) compared to SCC (6 patients, 30%). 18 patients had EGFR mutations. 10 patients had EFGR exon 19 deletion 9 (NM_005228.3:c.2239_2247del9), 2 patients had EFGR exon 19 deletion 15 (NM_005228.3:c.2235_2249del15 and c.2236_2250del15 - p.Glu746_Ala750del); 3 patients had EGFR exon 18 pont mutations (NM_005228.3 p.Gly719Ser, p.Gly719Cys, p.Gly719Ala, p.Gly719Asp) and 3 patients had EGFR somatic mutation c.2369C&gt;T (p.T790M) in exon 20 (NM_005228.5(EGFR):c.2369C&gt;T (p.Thr790Met)).

ALK rearrangements (positive ALK expression) were detected in 11 patients (29.7%), showing higher frequency in SCC (8 patients, 72.7%) than in ADC (3 patients, 27.3%).

ROS1 rearrangements (positive ROS-1 expression) were found in 12 patients (32.4%), equally distributed between ADC and SCC.

NTRK gene fusions (positive pan-TRK expression) were present in 3 patients (8.1%), with 2 cases in ADC and 1 in SCC, all in late-stage disease. The distribution of genetic biomarkers across cancer subtypes is shown in Figure 4.

Peritumoral inflammatory infiltration (PII) showed distinct patterns between cancer subtypes. ADC cases, particularly in late stages, demonstrated a higher tendency for inflammatory infiltration, with 90% showing some degree of infiltration (PII ≥ 1). In contrast, SCC cases exhibited more variable PII patterns, with 62.5% of late-stage cases showing no inflammatory infiltration (PII = 0) compared to 33.3% in early-stage cases.

A strong positive correlation was observed between FOXP3 and PD-L1 expression (r=0.76), consistent with previous findings in other cancer types. This relationship suggests potential interactions between these immune regulatory pathways in lung cancer progression.

### 3.3. Biophysical Properties Analysis

Biophysical measurements provided insights into the mechanical properties of lung cancer cells. Elasticity, quantified as Young’s modulus, exhibited distinct patterns across cancer stages and grades. Both nucleus and cytoplasm elasticity were found to have moderate positive correlations with cancer stage, with significantly stiffer cells observed in late-stage samples compared to early-stage. Additionally, cytoplasm elasticity showed a moderate negative correlation with tumor grade, suggesting that more poorly differentiated tumors may have more compliant cytoplasm.

Cell dimension measurements revealed distinct patterns between invasive and non-invasive samples, as illustrated in Figure 5. Comparison of cell heights and widths between invasive and non-invasive samples showed notable differences.

Surface roughness, measured as RMS roughness, demonstrated moderate positive correlations with cell dimensions, including nucleus height and cell width. This indicates that larger cells may exhibit rougher surfaces. A weak positive correlation was also found between surface roughness and cancer stage, hinting that more advanced cancers may have slightly rougher cell surfaces.

A representative example of surface roughness analysis is shown in Figure 6. The AFM height image (Figure 6a) shows a 20 × 20 $\mu m^{2}$ scan area with selected 5 × 5 $\mu m^{2}$ regions used for detailed roughness analysis. The corresponding measurements (Figure 6b) demonstrate how RMS roughness values were calculated from these selected regions, providing quantitative assessment of surface texture variations in cancer tissue samples.

Further details on the relationships between biophysical properties and clinical and histopathological parameters are provided in the subsequent section.

### 3.4. Correlations Between Biophysical and Clinical Parameters

Correlation analysis revealed several significant relationships between the biophysical measurements and conventional clinical indicators.

Both nucleus and cytoplasm elasticity exhibited moderate positive correlations with cancer stage (r = 0.621 and r = 0.444 respectively), suggesting that more advanced cancers tend to have stiffer cells. This trend was statistically significant, with early-stage samples showing significantly lower mean nucleus elasticity (1.98 ± 1.99 MPa) compared to late-stage samples (12.72 ± 7.26 MPa) (Mann-Whitney U test, *p* < 0.05). A similar pattern was observed for cytoplasm elasticity, with early-stage samples having a mean of 3.89 ± 4.30 MPa compared to 15.50 ± 12.46 MPa in late-stage samples (Mann-Whitney U test, *p* < 0.05). Figure 7 demonstrates the significant differences in cell elasticity between early and late stage samples.

Additionally, cytoplasm elasticity demonstrated a moderate negative correlation with tumor grade (r = −0.533), indicating that more poorly differentiated tumors may have more compliant cytoplasm.

Interestingly, nucleus elasticity was found to have a moderate positive correlation with vascular invasion (r = 0.475), while cytoplasm elasticity showed a moderate positive correlation with perineural invasion (r = 0.321). However, the lack of non-invasive samples in this study prevents a comparative analysis to fully evaluate the hypothesis that cell elasticity can serve as an indicator of cancer invasiveness.

Surface roughness measurements also exhibited interesting relationships. RMS roughness demonstrated moderate positive correlations with cell nucleus height (r = 0.343) and cell width (r = 0.321), suggesting that larger cells may have rougher surfaces. A weak positive correlation between RMS roughness and cancer stage (r = 0.258) was also observed, indicating that more advanced cancers may have slightly rougher cell surfaces.

### 3.5. Performance of Machine Learning Models

We applied various machine learning models to the LCPP dataset using Leave-One-Out Cross-Validation (LOOCV). Table 4 presents the performance metrics of the traditional machine learning models.

Table 4 demonstrated that Decision Trees and SVM models achieved the highest accuracy of 89.19%. Furthermore, Recursive Feature Elimination (RFE) consistently improved model performance, identifying four key features: sex, vascular invasion (VI), perineural invasion (PNI), and ALK mutation.

### 3.6. Bayesian Network Performance

We developed Bayesian Network (BN) models using two feature sets (with and without biophysical measurements) and two network structures (learned and inferred). Table 5 summarizes the performance of these models.

For our best performing Bayesian Network model with learned structure, from the confusion matrix we calculate:Cohen’s Kappa = 0.783, indicating substantial agreement beyond chanceMatthews Correlation Coefficient = 0.784, suggesting strong correlation between predicted and actual classes

The inclusion of biophysical features in the BN model with learned structure resulted in a slight improvement in accuracy from 86.49% to 89.19%. However, McNemar’s test yielded a *p*-value of 1.0, indicating no statistically significant difference in accuracy between the models with and without biophysical features. DeLong’s test for AUC comparison resulted in a *p*-value of 0.0, suggesting a significant difference in AUC, albeit with a slight decrease when biophysical features were added.

The ROC curves for the Bayesian Network models with learned and inferred structures are presented in Figure 8, illustrating the trade-off between true positive rate and false positive rate across various classification thresholds.

The performance of the Bayesian Network models with learned and inferred structures is visually represented in Figure 9, which shows the confusion matrices for both models.

### 3.7. Feature Importance and Model Interpretability

The BN model using only the four features identified by RFE achieved performance identical to the full-feature model, with a much lower Bayesian Information Criterion (BIC) of 51.931, compared to 268.586 for the full-feature model. This indicates a better trade-off between model fit and complexity. The learned BN structures provided insights into variable relationships:Without biophysical features: strong connections between cancer type, sex, and smoking history.With biophysical features: potential relationships between cell elasticity and cancer stage, and between surface roughness and cell dimensions.
These structures offer interpretable representations of the probabilistic relationships in the data, potentially informing future research directions.

## 4. Discussion

This study investigated the potential of integrating biophysical measurements with conventional clinical and histopathological tumour characteristics (tumour type, grade, invasiveness, stage) to enhance lung cancer subtype classification. Our findings provide several insights into the utility of this approach and its implications for future research in cancer diagnostics.

### 4.1. Integration of Biophysical Features

The inclusion of AFM-derived features improved classification accuracy from 86.49% to 89.19% when using a data-driven Bayesian Network model. While this improvement was not statistically significant (McNemar’s test, *p* = 1.0), the high Cohen’s Kappa (0.783) and Matthews Correlation Coefficient (0.784) suggest robust model performance. The decrease in AUC-ROC from 0.903 to 0.847 with biophysical features indicates a potential trade-off between accuracy and ranking performance that warrants further investigation with larger datasets.

These results suggest that biophysical properties of cancer cells may contain relevant information for subtype classification, aligning with previous studies that have demonstrated the potential of cell mechanical properties as biomarkers for cancer detection and characterization [7,8]. However, the limited improvement in our study indicates that the relationship between biophysical properties and cancer subtypes may be more complex than initially hypothesized.

### 4.2. Performance of Machine Learning Models

Our comparison of various machine learning techniques revealed that relatively simple models, such as Decision Trees and Support Vector Machines, performed well on our dataset. This finding is consistent with previous studies on lung cancer classification [4,9], suggesting that with appropriate feature selection, these models can effectively capture relevant patterns for subtype classification.

The strong performance of Bayesian Networks, particularly with learned structures, highlights their potential in handling complex, multidimensional medical data. The interpretability of Bayesian Networks, allowing for visualization of probabilistic relationships between variables, offers an advantage in clinical settings where understanding model decisions is crucial.

### 4.3. Feature Importance and Selection

The identification of four key features (sex, vascular invasion, perineural invasion, and ALK positive expression) through Recursive Feature Elimination is a significant finding. The fact that a model using only these features achieved performance comparable to the full-feature model suggests that a small set of carefully selected variables may be sufficient for accurate lung cancer subtype classification. This aligns with the principle of parsimony in model selection and could have practical implications for clinical applications, potentially reducing the need for extensive and costly tests.

The importance of sex and ALK mutation in our model is consistent with previous research highlighting gender differences in lung cancer incidence and the role of genetic alterations in cancer development [16,17]. The significance of vascular and perineural invasion in our model confirmed previous findings and extent by the monstration of the significant value of such histopathological features in cancer characterization [18,19].

### 4.4. Limitations and Future Directions

The primary limitation of this study is the small sample size (n = 37), which may have affected the statistical power of our analyses and the generalizability of our findings. Future studies with larger cohorts are needed to validate these results and potentially discover more significant associations between biophysical properties and cancer subtypes.

While our elasticity measurements showed strong statistical power, the comparison of classification accuracies with and without biophysical features was limited by low statistical power due to the small effect size (0.08) and sample size. This suggests that future studies with larger cohorts are needed to definitively establish the impact of biophysical measurements on classification accuracy. Despite this limitation, the strong statistical power of our elasticity measurements provides confidence in the biological significance of the observed mechanical differences between cancer stages.

The presence of missing data in the biophysical measurements is another limitation. While Bayesian Networks allowed us to handle this issue without imputation, developing more robust methods for dealing with missing data in small medical datasets remains an important area for future research.

Future studies could explore more advanced feature engineering techniques to potentially extract more informative representations of cellular characteristics from AFM data. Additionally, investigating the integration of other novel biomarkers or imaging modalities with biophysical measurements could provide a more comprehensive approach to cancer classification.

### 4.5. Implications for Cancer Research and Clinical Practice

Our study demonstrates the feasibility of incorporating AFM-derived biophysical measurements into lung cancer subtype classification. While the improvement in classification accuracy was modest, the approach opens up new directions for cancer research, potentially leading to a more comprehensive understanding of the relationship between cellular physical properties and cancer biology.

The interpretability of Bayesian Network models, as demonstrated in this study, could be particularly valuable in clinical settings. These models not only provide predictions but also offer insights into the relationships between different clinical, histopathological, and biophysical features, which could inform clinical decision-making and guide further diagnostic or therapeutic strategies.

## 5. Conclusions

To conclude, this study provides evidence for the potential utility of integrating biophysical measurements with conventional clinical and histopathological characteristics in lung cancer subtype classification. While the improvements in classification accuracy were modest, our findings highlight the complexity of this integration and suggest several promising directions for future research. Furthermore, by integrated of machine learning and AFM data with conventional clinical and histopathological characteristics could opened new vistas in the diagnostics, prognosis and personalized treatment of malignant tumours.

## Figures and Tables

**Figure 1 diagnostics-15-00127-f001:**
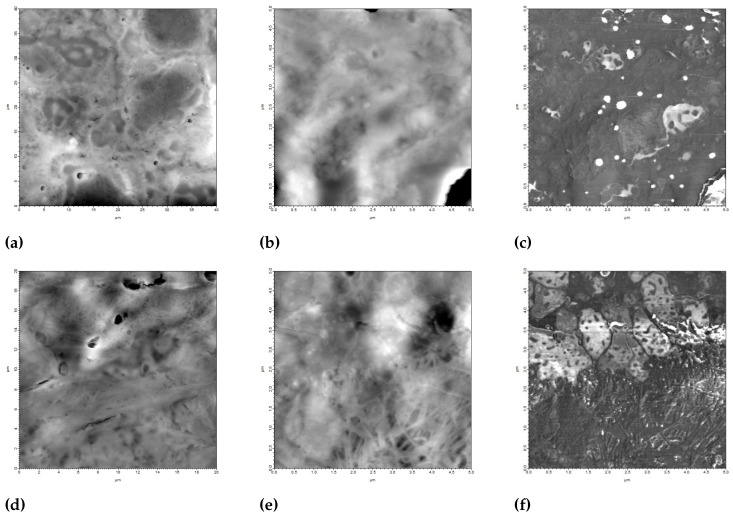
AFM measurement methodology. (**a**) AFM height image (40 × 40 $\mu m^{2}$) of lung adenocarcinoma tissue section; (**b**) Higher resolution AFM height image (5 × 5 $\mu m^{2}$); (**c**) Corresponding AFM phase image. Similar sequence of images (**d**–**f**) shows lung squamous cell carcinoma tissue section measurements.

**Figure 2 diagnostics-15-00127-f002:**
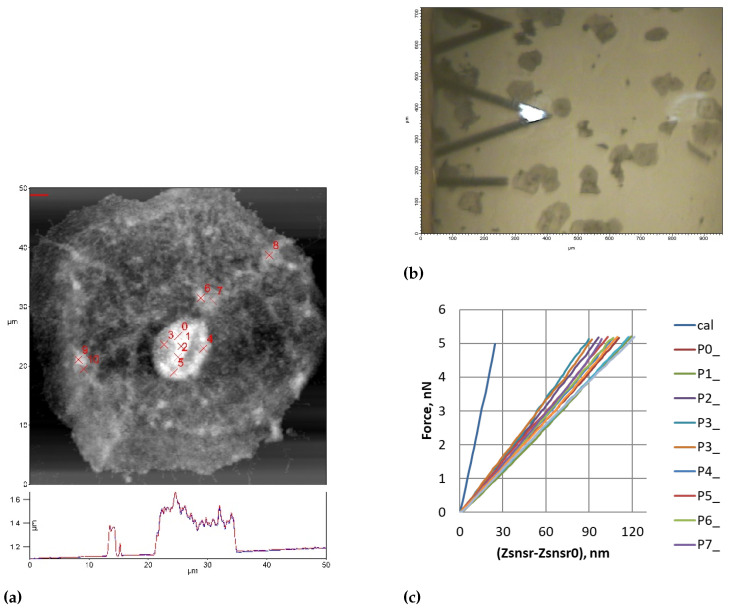
Example of force measurements on individual cancer cell. (**a**) AFM height image of individual cell with marked measurement points (P0–P7); (**b**) Optical image showing AFM probe and cells; (**c**) Force-displacement curves from cytoplasm and nucleus regions with calibration curve.

**Figure 3 diagnostics-15-00127-f003:**
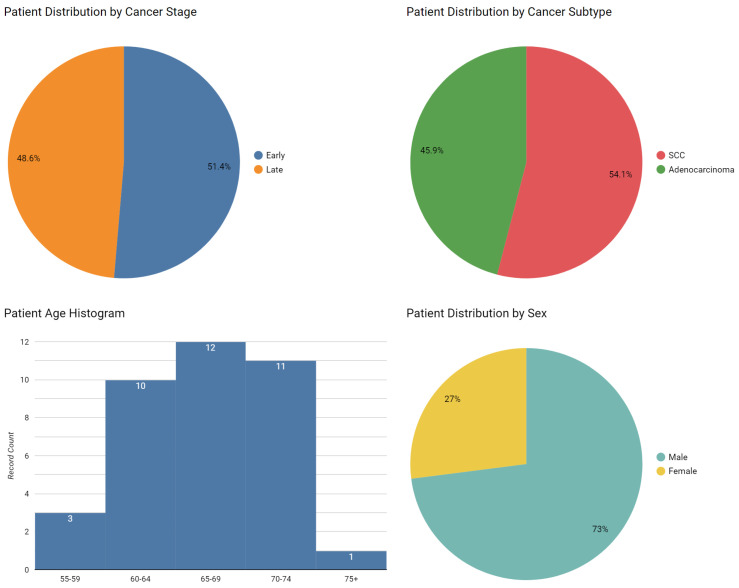
Demographic and clinical characteristics of the study cohort. Distribution of age by cancer subtype; Sex distribution; Smoking history in pack-years; Stage distribution by cancer subtype.

**Figure 4 diagnostics-15-00127-f004:**
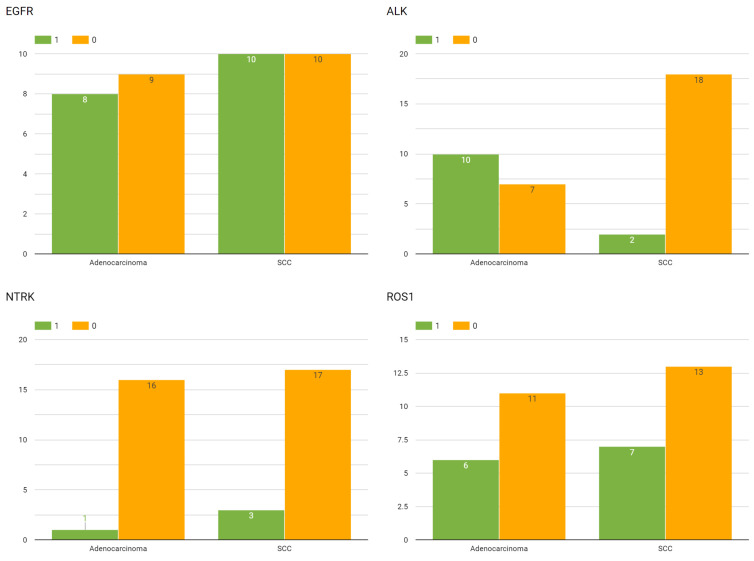
Distribution of genetic biomarkers by cancer subtype. Bar plot showing prevalence of EGFR, ALK, ROS1, and NTRK mutations in adenocarcinoma versus squamous cell carcinoma patients.

**Figure 5 diagnostics-15-00127-f005:**
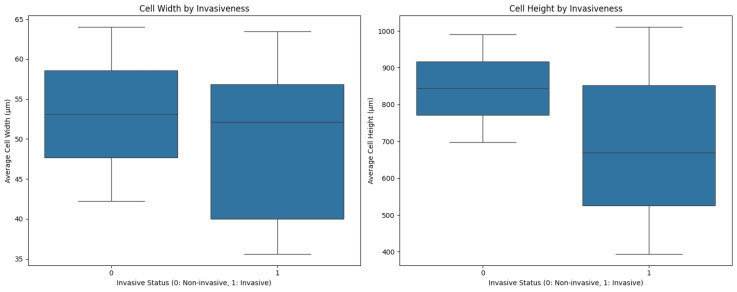
Cell dimension analysis by cancer invasiveness. Cell height measurements comparing invasive versus non-invasive samples; Cell width measurements comparing invasive versus non-invasive samples.

**Figure 6 diagnostics-15-00127-f006:**
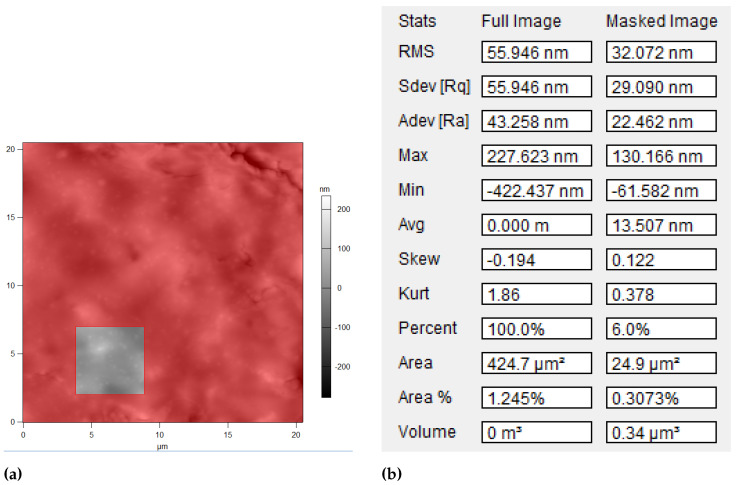
Surface roughness analysis. (**a**) Example of 20 × 20 $\mu m^{2}$ AFM height image with selected 5 × 5 $\mu m^{2}$ analysis regions; (**b**) Corresponding roughness measurements.

**Figure 7 diagnostics-15-00127-f007:**
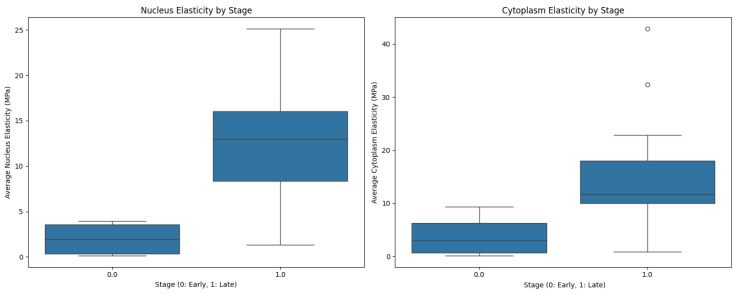
Cell elasticity measurements by cancer stage. Nucleus region elasticity comparison between early and late stage samples; Cytoplasm region elasticity comparison between early and late stage samples. Early stage samples show significantly lower elasticity in both regions (Mann-Whitney U test, *p* < 0.05).

**Figure 8 diagnostics-15-00127-f008:**
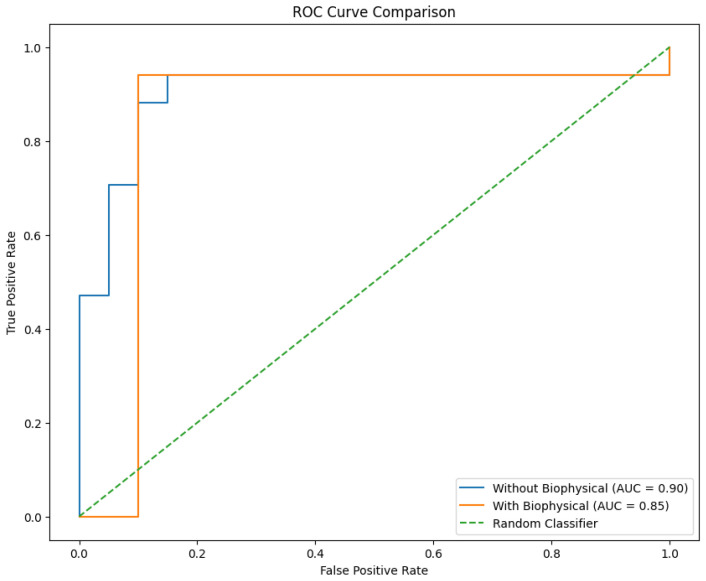
Receiver Operating Characteristic (ROC) curves for Bayesian Network models using all features. The blue curve represents the model with learned structure, while the orange curve represents the model with inferred structure. The area under each curve (AUC) provides a measure of the model’s overall performance.

**Figure 9 diagnostics-15-00127-f009:**
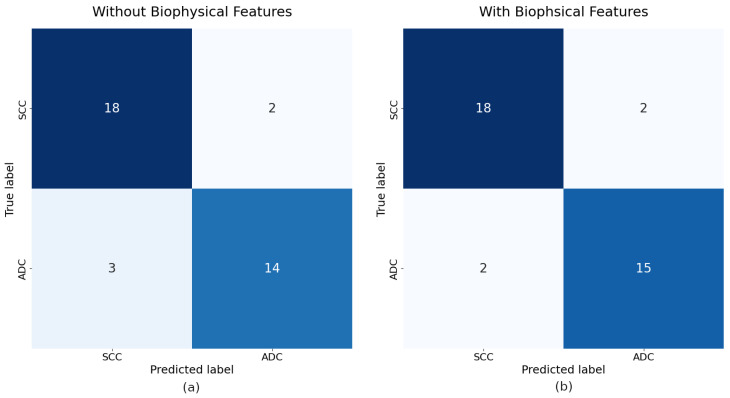
Confusion matrices for the Bayesian Network models using all features. (**a**) Model without biophysical features. (**b**) Model with biophysical features. The matrices illustrate the classification performance of each model in terms of true positives, true negatives, false positives, and false negatives.

**Table 1 diagnostics-15-00127-t001:** Overview of recent NSCLC classification studies—Part 1.

Study	Dataset Size	Features	Preprocessing
Liu et al. (2019) [2]	349 patients	1029 CT radiomics features	$\ell_{2,1}$-norm minimization
Sha et al. (2019) [3]	100 patients	857 PET radiomics features	Multi-step: variance filtering, LASSO
Shen et al. (2021) [4]	250 patients	Clinical data, 385 PET/CT radiomics	Boruta algorithm
Wang et al. (2022) [9]	138 patients	Clinical, laboratory, CT radiomics	LASSO with CV
Zhao et al. (2022) [10]	120 patients	95 mixed features	Boruta algorithm
Current Study	37 patients	Clinical, histopathological, AFM	RFE, percentile discretization

**Table 2 diagnostics-15-00127-t002:** Overview of recent NSCLC classification studies—Part 2.

Study	ML Method	Results
Liu et al. [2]	SVM	Acc: 86.00%
Sha et al. [3]	Logistic Regression	AUC: 0.781
Shen et al. [4]	SVM-RBF	Acc: 86.23%, AUC: 0.899
Wang et al. [9]	Random Forest	AUC: 0.995
Zhao et al. [10]	SVM	SVM: Acc 80.00%, AUC: 0.876
Current Study	Bayesian Networks	Acc: 89.19%, AUC: 0.847

**Table 3 diagnostics-15-00127-t003:** Demographic, clinical, histopathological, and genetic characteristics in the dataset.

Feature Name	Data Type	Encoding
Sex	categorical	1—female 2—male
Age, years	continuous numerical	
Pack years smoked	continuous numerical	
Cancer type	categorical	1—lung SCC 2—lung ADC
Cancer stage	categorical	0—early stage 1—late stage
Cancer Grade	categorical	1—well differentiated 2—moderately differentiated 3—poorly differentiated
Cancer size, cm	continuous numerical	
PD-L1, %	continuous numerical	
LVI	categorical	0—no lymphatic vessel invasion 1—lymphatic vessel invasion present
VI	categorical	0—no vascular invasion 1—vascular invasion present
PNI	categorical	0—no perineural invasion 1—perineural invasion present
PII	categorical	0—no peritumoral inflammation 1—mild inflammation 2—moderate inflammation 3—severe inflammation
FOXP3, cell count/mm^2^	continuous numerical	
EGFR gene mutation (NM_005228.3)	categorical	0—wild type 1—mutant
ROS1 expression	categorical	0—negative ROS-1 expression 1—positive ROS-1 expression
ALK expression	categorical	0—negative for ALK 1—positive for ALK
Pan-TRK expression	categorical	0—negative pan-TRK expression 1—positive pan-TRK expression

**Table 4 diagnostics-15-00127-t004:** Performance of Traditional Machine Learning Models.

Model	Accuracy (%)	Precision (%)	Sensitivity (%)	Specificity (%)	AUC
DT	89.19	88.24	88.24	90.00	0.871
SVM (RBF)	89.19	88.24	88.24	90.00	0.859
SVM (Polynomial)	89.19	88.24	88.24	90.00	0.847
LR	86.49	87.50	82.35	90.00	0.776
LDA	83.78	86.67	76.47	90.00	0.853
KNN	83.78	86.67	76.47	90.00	0.834

**Table 5 diagnostics-15-00127-t005:** Performance of Bayesian Network Models.

Features	Structure	Accuracy (%)	Precision (%)	Sensitivity (%)	Specificity (%)	AUC
Without Biophysical	Learned	86.49	86.53	82.35	90.00	0.903
All	Learned	89.19	89.19	88.24	90.00	0.847
Top 4	Learned	89.19	89.19	88.24	90.00	0.847
Without Biophysical	Inferred	72.97	72.97	70.59	75.00	0.744
All	Inferred	72.97	72.97	70.59	75.00	0.750

## Data Availability

The datasets presented in this article are not readily available because they contain sensitive medical information that could potentially lead to the identification of study participants, given the specific nature of the patient cohort (lung cancer patients from Latvia) and the inclusion of demographic data such as age and sex. Protecting patient privacy and confidentiality is paramount. Requests to access a de-identified version of the datasets should be directed to Andis Liepins (andis.liepins@applyit.lv), subject to approval and appropriate data-sharing agreements.

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
