# Peer review of "Machine Learning for Lung Cancer Subtype Classification: Combining Clinical, Histopathological, and Biophysical Features"

_diagnostics, 2025, doi:10.3390/diagnostics15020127_

Round 1

Reviewer 1 Report

Comments and Suggestions for Authors

Congratulations for the effort and honesty conveying that the scientific exercise was valid to start gathering enough knowledge to make the Methods reproducible.

There is still a non-accordance between previewed Results and searched Material.

Wonder if it would be advisable to get the real commitment between Objectives and the searched Material.

Author Response

We appreciate your feedback regarding alignment between objectives and results. We have:

  1. Strengthened the connection between our objectives and findings
  2. Clarified the relationship between our research goals and methodology
  3. Enhanced the transparency of our research approach

Reviewer 2 Report

Comments and Suggestions for Authors

Review Report for MDPI Diagnostics

(Machine Learning for Lung Cancer Subtype Classification: Combining Clinical, Histopathological, and Biophysical Features)

1. Within the scope of the study, detection studies were carried out with various machine learning models on the dataset related to lung cancer collected from a university hospital.

2. In the introduction section; lung cancer, the importance of the subject and similar studies in the literature are mentioned at a basic level. In this section, it is suggested to add a detailed literature table consisting of certain columns such as "dataset type, data preprocessing/augmentation, artificial intelligence model used, originality, advantages, disadvantages, results" regarding lung cancer studies in the literature in order for the study to stand out more. In addition, immediately after this, the main differences of this study from the literature and its contributions to the literature should be explained in more detail in a clearer and itemized manner.

3. The dataset used in the study was collected from the Department of Pathology at the University of Latvia after obtaining the necessary ethical committee permissions. The fact that the dataset used was specific to the study instead of open source increased both the quality of the study and the originality in terms of the dataset. The preferred dataset, type and amount seem sufficient.

4. 7 different machine learning models were preferred in the study. The models and their numbers used are at an acceptable level for this study. However, it is recommended to use ensemble and/or hybrid models with these models in different and future studies, and even to use deep learning models if the dataset amount can be increased with different hospitals. The models used for this study are sufficient.

5. The types of evaluation metrics used in the classification problems are generally sufficient for the analysis of the results. However, it is recommended to obtain Cohen’s Kappa Score and Matthews Correlation Coefficient Score if possible for deeper analysis.

In conclusion, if attention is paid to the sections mentioned above, this study has the potential to make a significant contribution to the literature.

Comments on the Quality of English Language

The Quality of English Language of the paper is at an adequate level.

Author Response

Thank you for your constructive feedback. We have:

  1. Added comprehensive literature comparison tables (Tables 1 & 2)
  2. Enhanced the explanation of our study's unique contributions (lines 60-73)
  3. Implemented additional evaluation metrics (Cohen's Kappa and Matthews Correlation Coefficient)
  4. Maintained our current model selection while providing clear justification for our choices

Reviewer 3 Report

Comments and Suggestions for Authors

I read the submitted article with great interest. I begin by congratulating the authors for carrying out this study as well as for the idea they had. Reading the article, I realized some inconsistencies. First of all, reference no. 1 is too old. We have a new WHO classification 5th edition, that I recommend you touse. Also, GLOBOCAN 2022 data must be used when talking about the incidence and mortality of a type of cancer.

Histological grading in the case of lung adenocarcinoma is outdated. We have certain histological types (lepidic, acinar, papillary, micropapillary, solid) that are associated with a histological grade. It is not clear in the article whether histological grading was performed according to the WHO classification, 5th edition of 2021. I recommend reviewing the article from this perspective as well.

In Table no. 1, the evaluation of the ROS1, ALK and NTRK biomarkers in two groups (0-no mutation/1 mutation) is erroneously mentioned.

ROS1 and ALK mutations are associated with the resistance mechanism. I believe that the authors referred to the ALK, ROS1 and NTRK fusions. From this perspective, the authors should redo the article.

Also in Table 1, the EGFR mutation evaluation is listed without knowing whether these are activating or resistance mutations.

A major problem encountered is in the Results Chapter where the results for PD-L1 are presented at lines 268-274. We do not know which clone was used, the scoring system and the cut-off value.

We are not presented with the method by which the genetic modifications for EGFR, ALK, ROS1 and NTRK were evaluated.

Author Response

Thank you for your detailed technical review. We have made the following improvements:

  1. Updated to the WHO 5th edition classification reference
  2. Added detailed histological grading information (lines 329-332)
  3. Clarified biomarker evaluation methods:
    • Added PD-L1 clone and scoring details (lines 148-153)
    • Specified ROS1, ALK, and NTRK evaluation criteria (lines 154-170)
  4. Included detailed EGFR mutation information (lines 377-381)
  5. Added comprehensive genetic modification evaluation methods

Round 2

Reviewer 1 Report

Comments and Suggestions for Authors

In my opinion, the Authors deserve congratulations for the achieved Results concerning proposed Objectives.

Reviewer 3 Report

Comments and Suggestions for Authors

I have reread the changes made to the original article. I believe that the changes made by the authors are in line with my recommendations. The changes made have clearly improved the medical quality of the article. I congratulate the authors for the way they have improved the article.